# Free GPIs and Comparison of GPI Structures Among Species

**DOI:** 10.3390/ijms262311592

**Published:** 2025-11-29

**Authors:** Stella Amarachi Ihim, Morihisa Fujita

**Affiliations:** Institute for Glyco-core Research (iGCORE), Gifu University, Gifu 501-1193, Japan; ihim.stella.amarachi.t9@f.gifu-u.ac.jp

**Keywords:** free GPI, GPI-anchored protein, mammalian cells, biosynthesis, *Trypanosoma brucei*, *Trypanosoma cruzi*, *Toxoplasma gondii*, *Plasmodium falciparum*, *Leishmania* spp.

## Abstract

Glycosylphosphatidylinositols (GPIs) are complex glycolipids that function as membrane anchors for a wide array of eukaryotic proteins, collectively referred to as GPI-anchored proteins (GPI-APs). These structures are critical for various cellular processes including signal transduction, host–pathogen interactions, and immune evasion. While GPI-APs have been extensively studied, increasing attention is being paid to non-protein-linked GPI, called free GPIs, which have been identified in both protozoan parasites and mammalian cells. In protozoa such as *Trypanosoma brucei*, *Trypanosoma cruzi*, *Toxoplasma gondii*, *Plasmodium falciparum*, and *Leishmania* spp., free GPIs play roles in virulence, immune modulation, and parasite survival. In mammals, free GPIs have been detected in several tissues and pathogenic conditions of paroxysmal nocturnal hemoglobinuria caused by PIGT mutation and rare blood group phenotypes. This review provides a comparative overview of the structure and biosynthesis of free GPIs and GPI-APs across species, highlighting unique adaptations in each. We also discuss the emerging physiological and pathological roles of free GPIs, proposing that these underexplored molecules may serve as important biomarkers and therapeutic targets. Understanding the diversity and function of free GPIs offers new insights into glycobiology and host–pathogen interactions.

## 1. Introduction

Glycosylphosphatidylinositols (GPIs) are glycolipids that anchor proteins to the outer leaflet of the plasma membrane in eukaryotic cells. This post-translational modification is essential for the surface localization of proteins that lack transmembrane domains, thereby contributing to diverse biological processes, including cell adhesion, signal transduction, and host cell defense [1]. The structure of GPI is conserved across species, comprising a core glycan of three mannoses (Man) linked to glucosamine (GlcN), which is attached to an inositol-phospho-lipid. Ethanolamine phosphate (EtNP) bridges link the glycan to the C-terminus of proteins. The GPI core structure can be represented as: EtNP-6-Man-α1,2-Man-α1,6-Man-α1,4-GlcN-α1,6-myo-Inositol-phospholipid (Figure 1A) [1]. More than 150 GPI-anchored proteins (GPI-APs) have been identified in mammals, where they serve as hydrolytic enzymes, receptors, adhesion molecules, and complement regulators. GPIs are also abundant in protozoan parasites such as *Trypanosoma brucei*, *T. cruzi*, *Leishmania* spp., *Toxoplasma gondii*, and *Plasmodium falciparum* (Figure 2), often forming dense surface coats that are critical for pathogenicity and immune evasion [2]. Beyond protein-linked GPIs, free GPIs—those not attached to proteins—have been discovered in protozoan parasites and, more recently, in mammalian cells [3,4]. In protozoa, free GPIs may be structurally like protein-linked GPIs or may possess distinct features. They exist in large quantities on the cell surface and can modulate host immune responses. In mammals, free GPIs appear to be byproducts of biosynthesis or regulated intermediates, with emerging evidence suggesting functional roles in metabolism, immunity, and disease.

This review provides a comparative analysis of the structure and biosynthesis of GPI and GPI-APs in mammals and five representative protozoan parasites. For each species, we also examine the current understanding of the roles of free GPIs, with particular emphasis on their potential physiological and pathological functions. Through this comparative approach, we aim to illuminate the evolutionary adaptations of GPI biology and explore how free GPIs contribute to health and disease.

## 2. Structure and Biosynthesis of GPI and GPI-APs in Mammals

### 2.1. Structure of GPI and GPI-APs in Mammals

In mammalian cells, a conserved GPI core structure can be modified with a fourth α1,2-Man and a glycan side-chain consisting of Neu5Ac-α2,3-Gal-α1,3-GalNAc-β1,4- (where Neu5Ac, Gal, and GalNAc are N-acetylneuraminic acid, galactose, and N-acetylgalactosamine, respectively) (Figure 1B). The lipid moiety in mammalian GPI is either 1-alkyl-2-acyl-glycerol or diacylglycerol [5]. The lipid moiety is remodeled in the Golgi to contain saturated fatty acids, especially stearic acid, at the sn-2 position. This remodeling enhances GPI-AP association with lipid rafts, which are implicated in signaling and trafficking. On the other hand, in erythrocytes, the 2-position of the inositol in GPI remains acylated, and GPI fatty acid remodeling does not occur [5].

### 2.2. Biosynthesis of GPI-APs in Mammals

The biosynthetic pathway of GPIs has been well documented in mammals. The biosynthetic process is a sequence of 11 reactions that begins on the cytoplasmic side of the endoplasmic reticulum (ER) membrane (Figure 3A) [6]. An N-acetylglucosamine (GlcNAc) from uridine diphosphate (UDP)-GlcNAc is transferred to the 6-position of inositol on PI by GPI-GlcNAc transferase to generate GlcNAc-PI. The GlcNAc-PI is then *N*-deacetylated to form GlcN-PI, followed by flip to the luminal side of the ER membrane. Subsequently, the 2-position of the inositol ring of GlcN-PI is acylated to generate GlcN-(acyl)PI. During the initial steps of GPI biosynthesis, GPI intermediates bearing 1-alkyl-2-acylglycerol in their lipid moiety are enriched. Three to four Man and three EtNP residues are then sequentially added to form the GPI complete precursor H8, a structure just before the attachment to proteins. The GPI complete precursor is transferred to proteins having a GPI-attachment signal, which is mediated by a GPI transamidase complex (Figure 3A and Appendix A). Following GPI attachment to protein, an acyl-chain on inositol and a side-chain EtNP linked to the second Man (Man2) are removed from GPI-anchors in the ER, and the remodeled GPI-APs are transported to the Golgi apparatus. At the Golgi apparatus, GPI fatty acid remodeling is first carried out, in which an unsaturated fatty acid at the sn-2 position on the GPI lipid moiety is removed and replaced with a saturated fatty acid, usually a stearic acid. A glycan side-chain consisting of Neu5Ac-α2,3-Gal-α1,3-GalNAc-β1,4- modified with the first Man (Man1) on GPI-anchors is formed at the Golgi apparatus. Such remodeling reactions do not occur completely in all GPI-anchors and may terminate partially, resulting in microheterogeneity in GPI structures.

### 2.3. Free GPIs in Mammals

It is believed that free GPIs also undergo most of the biosynthetic processes except attachment to proteins [7]. Mammalian free GPIs are probably expressed on the cell surface when excess amounts of GPIs are synthesized. The presence of free GPIs, such as fully assembled GPI precursor H8 was detected on the cell surface of several mammalian cultured cell lines (Figure 1C) [7]. The cell surface expression of free GPIs seems to be temperature- and brefeldin A-sensitive, indicating vesicle-mediated transport.

Recently, another type of free GPI has been detected in cultured mammalian cells such as CHO-K1 cells and Neuro2a cells, and several tissues, including the mouse pons, medulla oblongata, spinal cord, testis, epididymis, and kidney. A monoclonal antibody, T5-4E10 (T5), which was originally isolated as an antibody against *T. gondii* free GPI from infected mice, could be used to detect such mammalian free GPIs bearing a GalNAc side-chain on Man1 as well (Figure 1C) [6]. Interestingly, at least fractions of the free GPIs are modified similarly to GPI-APs, including inositol deacylation and fatty acid remodeling, suggesting that free GPIs may follow the same biosynthetic and trafficking pathways. In HEK293 cells, even in the absence of GPI-transamidation, such as in PIGS-knockout (KO) cells, T5-positive free GPI is not detected on the cell surface [8]. However, when B3GALT4, which encodes the enzyme that adds Gal to the GalNAc residue of the GPI side-chain, is knocked out, T5-positive free GPI becomes detectable, suggesting that the side-chain GalNAc of GPI is normally masked by a Gal residue in HEK293 cells. On the other hand, defects in ER-associated degradation (ERAD) together with GPI transamidation in HEK293 cells lead to an increase in T5-positive free GPI [8,9]. This is because defects in ERAD cause the accumulation of precursor forms of specific GPI-APs, such as CD55, CD48, and PLET1, which in turn enhance GPI biosynthesis, leading to the appearance of free GPI on the cell surface.

The Emm antigen is a high-frequency red blood cell antigen that consists of free GPI carrying the Man2-linked EtNP side-chain [10] on the cell surface. Individuals who lack this antigen are extremely rare and are referred to as Emm-negative. A defect in the PIGG gene, which encodes an enzyme required for the transfer of the second EtNP to Man2 on GPI, results in the absence of Emm antigen on red blood cells, leading the immune system to produce anti-Emm antibodies against the missing GPI structure.

## 3. Structure and Biosynthesis of GPI and GPI-APs in *Trypanosoma brucei*

### 3.1. Life Cycle of T. brucei and Disease

*T. brucei* is a protozoan parasite that causes African trypanosomiasis (sleeping sickness) and animal trypanosomiasis (Nagana) in sub-Saharan Africa. The parasite undergoes two main proliferative stages: the bloodstream form in the mammalian host and the procyclic form in the tsetse fly vector. Transmission occurs via tsetse fly bites, introducing bloodstream trypomastigotes into the host. The parasite evades host immunity through antigenic variation in variant surface glycoproteins (VSGs), which form a dense, GPI-anchored coat on its surface. Approximately 1 × 10^7^ copies of the VSG are expressed on the surface of bloodstream-form parasites, accounting for ~10% of total protein [11,12]. The bloodstream forms of *T. brucei* require iron for growth, which is delivered by the transferrin of the host. The transferrin receptor of *T. brucei* is known to be different in structure, subunit organization, and mode of membrane anchorage from that of the mammalian host [13]. The trypanosome transferrin receptor, which is a heterodimer of related proteins ESAG6 and ESAG7, is attached to the cell membrane by a single GPI anchor on ESAG6 [14]. In the tsetse fly, VSGs are replaced by procyclins, also GPI-anchored. These GPI-APs are critical for parasite survival, transmission, and immune evasion.

### 3.2. Structure of GPI and GPI-APs in T. brucei

The GPI anchors in *T. brucei* share a conserved core with mammalian GPIs but have distinct modifications. In bloodstream form, VSG GPIs possess Gal side-chains (Figure 2A). The lipid moiety is a diacylglycerol containing two myristic acids. On the other hand, in procyclic form, procyclin GPIs contain sialylated poly-N-acetyllactosamine (Gal-β1,4-GlcNAc) and poly-lacto-N-biose (Gal-β1,3-GlcNAc) extensions, and the lipid moiety is a lyso-form (monoacylglycerol) with an additional acyl group on inositol. Free GPIs are expressed in both the bloodstream and procyclic forms [15,16]. Particularly, in procyclin-null mutant cells, free GPIs are abundantly produced. The structures seem to consist of a lower number of poly-N-acetyllactosamine and poly-lacto-N-biose repeats, compared to procyclin anchor, and a proportion contained diacylglycerol.

### 3.3. Biosynthetic Differences in T. brucei

The GPI biosynthetic pathway in *T. brucei* is similar in core steps to that of mammals. Gene knockout studies have revealed that when GPI biosynthesis is impaired (e.g., deletion of TbGPI12 or TbGPI10), GPI intermediates accumulate while GPI-APs are absent [15,17,18]. In bloodstream form *T. brucei*, TbGPI10 is essential, and its loss is lethal because GPI-anchored VSGs are required for survival, and probably in addition to the GPI-anchored transferrin receptor required for acquiring iron and viability. In contrast, procyclic forms can survive without TbGPI10, but they grow slowly in vitro and show abnormal adhesion. The TbGPI12 mutants can survive in culture, but fail to colonize the tsetse fly midgut, underscoring the importance of GPI anchoring for transmission [19]. There are key differences in acylation timing, fatty acid remodeling, and side-chain additions.

#### 3.3.1. Acylation Timing

In mammals, inositol acylation occurs to GlcN-PI and remains through protein attachment. In *T. brucei*, inositol acylation is initially added to Man-GlcN-PI and can be reversed before protein transfer (Figure 3B and Appendix A). In mammals, PIGW is responsible for the inositol acylation, whereas PGAP1 mediates inositol deacylation [20,21]. In *T. brucei*, there is no homolog of PIGW, suggesting that a different family enzyme is involved in the step. For GPI inositol deacylation in *T. brucei*, two GPI inositol deacylase, GPIdeAc and GPIdeAc2 (homolog of PGAP1) are reported [22,23]. In the bloodstream form, unlike the procyclic forms, GPI inositol-deacylation is essential to complete fatty acid remodeling.

#### 3.3.2. Fatty Acid Remodeling

Fatty acid remodeling in *T. brucei* occurs before protein attachment in the ER. The blood stream GPIs are remodeled to contain myristic acid (C14:0) at both sn-1 and sn-2 positions [24]. In the procyclic form, myristate exchange does not occur; instead, an inositol-acylated lyso-GPI is attached to the protein. In mammals, fatty acid remodeling at the sn-2 position of GPI lipid is carried out in the Golgi apparatus: the removal of an unsaturated fatty acid from sn-2 position of GPI lipid is mediated by PGAP3, and PGAP2 is involved in reacylation of a saturated stearic acid (Figure 3A) [25]. Recently, in *T. brucei*, Tb927.7.6110 was identified as the gene responsible for GPI-phospholipase A_2_ (GPI-PLA_2_) activity required for fatty acid remodeling at the sn-2 position of GPI precursors [26]. Tb927.7.6110 shows sequence similarity to mammalian PGAP6, a GPI-PLA_2_ that acts on the cell surface and is related to the PGAP3 family of proteins [27]. Disruption of Tb927.7.6110 impaired fatty acid remodeling and resulted in a reduction in GPI glycan side-chain structures, whereas reintroduction of the gene restored normal GPI architecture. Two related genes, Tb927.7.6170 and Tb927.7.6150, are located adjacent to Tb927.7.6110 in the *T. brucei* genome. Although reintroduction of Tb927.7.6170 led to partial restoration, biochemical analyses indicated that it lacks GPI-PLA_2_ activity toward GPI precursors. Tb927.7.6150 is presumed to be catalytically inactive due to substitutions in conserved residues within the predicted active site. Reacylation with myristic acid at sn-2 position of GPI lipid is mediated by TbGUP1, a homolog of yeast GUP1, a membrane-bound acyltransferase required for GPI fatty acid remodeling [28]. The responsible genes involved in fatty acid remodeling at sn-1 position of GPI lipid in *T. brucei* are elusive. In addition to fatty acid remodeling, *T. brucei* performs myristate exchange, a post-translational reaction in which the myristic acids of GPI-anchored VSG are replaced with new myristate molecules [29]. This process occurs after the ER, likely in endosomal compartments during VSG recycling, and serves as a proofreading or repair mechanism to ensure that all GPI anchors remain fully myristoylated.

#### 3.3.3. Side-Chain Addition

In mammalian GPI biosynthesis, EtNP groups are added to all three mannose residues (Man1–3) of the glycan core (Figure 3A). Among them, the third EtNP, known as the bridging EtNP, is used to link the GPI anchor to the protein, while the first EtNP is always present. The second EtNP attached to Man2 is removed after GPI is transferred to proteins in mammalian cells [30]. In contrast, in protozoa, including *T. brucei*, only the bridging EtNP required for protein attachment is added, and side-chain EtNPs are absent (Figure 3B). Furthermore, in procyclic form of *T. brucei*, GPI anchors are modified with glycan side-chains composed of poly-LacNAc and poly-lacto-N-biose chains. These side-chains are capped by sialic acid residues, whereas in bloodstream forms, side-chains are exclusively Gal-based. Several glycosyltransferase genes have been implicated in building these side-chains. TbGT3 and TbGT8 are β1,3-Gal transferase and a β1,3-GlcNAc transferase, respectively, that are required for the synthesis of branched LacNAc units in both procyclic GPI-glycans and N-glycans [31]. Meanwhile, TbGT10 is a β1,6-GlcNAc transferase required for both procyclic GPI-glycans and bloodstream N-glycans [31]. Sialic acid capping on the poly-LacNAc and in procyclic form of *T. brucei*, GPI anchors are modified with glycan side-chains composed of poly-LacNAc and poly-lacto-N-biose structure is mediated by GPI-anchored trans-sialidase on the cell surface of the procyclic form [32,33]. This enzyme transfers sialic acid from host donor glycoconjugates onto terminal β-Gal residues of the GPI glycan side chains, forming an α2,3-linked sialic acid.

## 4. Structure and Biosynthesis of GPI and GPI-APs in *Trypanosoma cruzi*

### 4.1. Life Cycle of T. cruzi and Disease

*T. cruzi* is the causative agent of Chagas disease, endemic in Latin America and increasingly observed worldwide. Transmission typically occurs via triatomine insects, but congenital, oral, and transfusion-based transmission are also possible. The life cycle includes Epimastigotes: non-infectious forms proliferating in the insect midgut; Metacyclic trypomastigotes: infective forms in the insect hindgut, transmitted to humans via fecal contamination; Amastigotes: intracellular replicative forms in mammalian cells; and Bloodstream trypomastigotes: non-dividing, extracellular forms responsible for dissemination.

### 4.2. Structure of GPI and GPI-APs in T. cruzi

The surface of *T. cruzi* is densely packed with mucin-like glycoproteins anchored via GPI. These molecules are essential for parasite protection and host invasion. The lipid moiety of many GPIs in *T. cruzi* consists of ceramide [34]. Free GPIs in *T. cruzi*, formerly referred to as lipopeptidophosphoglycan (LPPG), are called glycoinositolphospholipids (GIPLs), which are major components of the parasite surface and contribute to immune evasion and host–pathogen interactions (Figure 2B) [35]. The estimated copy number of GIPLs is approximately 1.5 × 10^7^ molecules per cell [36], whereas ~2–4 × 10^6^ copies of GPI-anchored mucins are expressed per parasite [37]. Structural features include Ceramide-based lipids, a conserved Man_4_-GlcN-inositol-phospholipid core, modifications with EtNP or 2-aminoethylphosphonic acid (AEP), and glycan side-chains including β-galactofuranose units. The glycan and lipid heterogeneity of *T. cruzi* GPIs reflect its strain diversity and may influence infectivity, immune recognition, and host cell signaling. The abundance of free GPI at all life cycle stages suggests that these molecules are not simply biosynthetic intermediates but may play active roles in parasite survival, host immune modulation, and metabolic interference.

## 5. Structure and Biosynthesis of GPI and GPI-APs in *Toxoplasma gondii*

### 5.1. Life Cycle of T. gondii and Disease

*T. gondii* is an obligate intracellular parasite capable of infecting nearly all warm-blooded animals, including humans. It is the causative agent of toxoplasmosis, which poses serious risks to immunocompromised individuals and fetuses [38]. Its life cycle includes three infectious forms: Tachyzoites, rapidly dividing forms responsible for acute infection; Bradyzoites, slowly dividing, encysted forms that persist in tissues during chronic infection; and Sporozoites, forms found within oocysts shed by felid definitive hosts. Tachyzoites actively invade host cells and express a dense coat of GPI-APs and free GPIs that are critical for host cell interaction and immune modulation.

### 5.2. Structure of GPI and GPI-APs in T. gondii

*T. gondii* tachyzoites actively invade host cells and express a dense coat of GPI-APs and free GPIs that are critical for host cell interaction and immune modulation. Free GPIs are present at high levels (~1 × 10^6^ copies per cell) and have been detected using specific monoclonal antibodies (e.g., T3-3F12 and T5-4E10) [4]. Both GPI-APs and free GPIs share a conserved glycan core: Man-α1,2-Man-α1,6-Man-α1,4-GlcN-α1,6-Inositol-phospholipid. In addition, a major GPI glycoform found in tachyzoites carries a distinctive side-chain consisting of Glc-α1,4-GalNAc-β1,4- (Glc, glucose) from Man1 (Figure 2C). This structure is highly immunogenic, eliciting strong IgM responses in humans and serving as a potential diagnostic marker. Another GPI variant lacks the Glc moiety and functions as a membrane anchor.

### 5.3. Biosynthetic Differences in T. gondii

While the core GPI biosynthetic steps in *T. gondii* resemble those in other eukaryotes, the parasite exhibits several distinctive features. The branching Glc-α1,4-GalNAc-β1,4-disaccharide side-chain is unique to *T. gondii* [39]. The addition of this β1,4-GalNAc and α1,4-Glc is mediated by glycosyltransferases specific to *T. gondii*, PIGJ, and PIGE, respectively (Figure 3C and Appendix A) [40]. Deletion of PIGJ results in GPIs lacking the side-chain branch and paradoxically enhances parasite virulence, suggesting that the side-chain may attenuate host immune responses [39]. Virulent RH strains and avirulent type II strains (e.g., PTG) both express free GPIs, but their quantitative content, glycosylation patterns differ while their immune reactivity is largely similar [41]. These differences likely affect how each strain interacts with host immune cells. Functional studies have shown that free GPIs from *T. gondii* activate TLR2/TLR4/MyD88-dependent signaling in macrophages, leading to TNF-α and IL-12 production [42,43]. This activation is modulated by galectin-3, which binds the unique sugar moieties of *T. gondii* GPIs [44]. Such interactions contribute to the parasite’s immunomodulatory capacity.

Importantly, GPI-AP-deficient mutants still produce free GPIs, suggesting that these molecules are not merely biosynthetic byproducts but have distinct biological functions. The abundance and immunogenicity of free GPIs make them promising targets for diagnostic assays or vaccine design.

## 6. Structure and Biosynthesis of GPI and GPI-APs in *Plasmodium falciparum*

### 6.1. Life Cycle of P. falciparum and Disease

*Plasmodium falciparum* is the most virulent species of the malaria-causing parasites in humans. Its complex life cycle involves two hosts: the Anopheles mosquito and humans. In humans, the parasite infects hepatocytes and subsequently erythrocytes, leading to the cyclical fevers and pathology characteristic of malaria. Severe disease results from the sequestration of infected red blood cells and excessive inflammatory responses. A hallmark of *P. falciparum*–infected erythrocytes is the presence of parasite-derived GPI-APs and free GPIs on the cell surface, which serve as potent immune stimuli and contribute to pathogenesis [45]. The merozoites are responsible for the invasion of red blood cells in the human hosts.

### 6.2. Structure of GPI and GPI-APs in P. falciparum

*P. falciparum* GPIs share the conserved Man_4_-GlcN-inositol core structure. There are both GPI-APs, such as merozoite surface protein 1 (MSP1) and MSP2, and free GPIs, which exist in greater abundance and are highly immunogenic. Free GPIs in *P. falciparum* are typically tetramannosylated (Figure 2D). The lipid portion retains the original fatty acids used during biosynthesis. These GPIs are resistant to PI-PLC but sensitive to GPI-specific phospholipase D (GPI-PLD), confirming their structural distinction from mammalian GPIs. Importantly, studies show that free GPIs are four to five times more abundant than protein-linked GPIs in the parasite, and the host’s immune response—particularly IgG—targets the free forms more strongly [45,46,47,48].

### 6.3. Biosynthetic Differences in P. falciparum

GPI biosynthesis in *P. falciparum* retains the general eukaryotic features but with several parasite-specific differences. As in other protozoan parasites, in Plasmodium, the addition of EtNP occurs only on Man3, which is required for protein attachment, and no side-chain EtNPs are added. Unlike mammalian GPIs, which undergo remodeling, *P. falciparum* GPIs retain an acyl-chain linked to inositol, likely due to the absence of inositol deacylase (Figure 3D and Appendix A). In addition, *P. falciparum* has a rudimentary Golgi apparatus, and post-ER processing of glycoproteins and glycolipids is limited [46]. As a result, there is no remodeling in the Golgi, unlike mammalian cells.

Functionally, *P. falciparum* GPIs act as potent pathogen-associated molecular patterns (PAMPs), activating innate immune receptors (e.g., TLR2 and TLR4) and stimulating production of pro-inflammatory cytokines such as TNF-α and IL-1β [49,50]. This excessive cytokine release is linked to the symptoms of malaria, including fever, hypoglycemia, and cerebral complications.

## 7. Structure and Biosynthesis of GPI and GPI-APs in *Leishmania* spp.

### 7.1. Life Cycle of Leishmania and Disease

*Leishmania* spp. are protozoan parasites responsible for a spectrum of diseases collectively known as leishmaniasis, which includes cutaneous, mucocutaneous, and visceral forms [51]. The life cycle alternates between: Promastigotes: flagellated forms in the gut of the sandfly vector, and Amastigotes: intracellular forms residing within phagolysosomes of mammalian macrophages.

During transmission, sandfly-injected promastigotes are phagocytosed by host macrophages and differentiate into amastigotes, which are adapted to survive in the hostile intracellular environment. The surface of both life stages is rich in GPI-anchored molecules and free GPIs, which form a protective glycocalyx and play critical roles in virulence and immune evasion.

### 7.2. Structure of GPI and GPI-APs in Leishmania

*Leishmania* produces a diverse array of GPI-related molecules including: GPI-APs (e.g., gp63 metalloprotease), Lipophosphoglycan (LPG): a highly glycosylated, GPI-anchored phosphoglycan, glycoinositolphospholipids (GIPLs): free GPIs not linked to protein or polysaccharide (Figure 2E) [52,53]. GIPLs and LPG are present at approximately 1 × 10^7^ and ~5 × 10^6^ molecules per cell, respectively [54]. All share a common Man-α1,4-GlcN-inositol-phospholipid core, but differ in their glycan headgroups and lipid moieties. The lipid moiety is either an alkyl-acyl-glycerol or a mono-alkylglycerol.

LPG is the dominant surface component of promastigotes. It consists of a lyso-alkyl-PI with a Gal-α1,6-Gal-α1,3-Galf-β1,2-(Glc-α1-phosphate-6-)Man-α1,3-Man-α1,4-GlcN-α1,6 core glycan, a linear phosphoglycan (PG) chain with repeating Gal-β1,4-Man-α1-phophate-6- unit, and a terminal oligosaccharide cap with species- and strain-specific modifications (Figure 2E). LPG helps *Leishmania* resist complement lysis, adhere to macrophages, and modulate phagolysosome maturation [55]. LPG is downregulated in amastigotes.

GIPLs are simpler in structure to LPG. They are classified into Type-1, Type-2, and Hybrid GIPLs. Type-1 GIPLs share similarity with protein-linked GPIs; Man2 linked at the C6 position of Man-GlcN-PI. Type-2 GIPLs resemble LPG cores, which possess Man2 linked at the C3 position of Man-GlcN-PI. Hybrid GIPLs feature Man branches at both C3 and C6 positions of Man-GlcN-PI. The lipid moiety of GIPLs varies, typically containing alkyl-acyl phosphatidylinositol (PI) with saturated fatty acids. Type-1 GIPLs often contain C14:0, while Type-2 show greater variability (C18:0 to C26:0). This heterogeneity might impact their immunogenicity. For example, GIPLs from *L. braziliensis* (galactose-rich, Type II) were shown to more strongly inhibit nitric oxide and IL-12 production in primed macrophages compared with GIPLs from *L. infantum* (Type I/hybrid) [51], indicating that structural differences can modulate macrophage responses.

### 7.3. Biosynthetic Differences in Leishmania

The biosynthesis of GPI-APs, LPG, and free GPIs in *Leishmania* involves shared early steps, but diverges at glycan elongation and side-chain modification (Figure 3E and Appendix A). All GPI-related molecules are derived from a common intermediate, Man-GlcN-PI, formed in the ER. Specific mannosyltransferase generates α1,3-linked Man branches on intermediates leading to Type-2 GIPLs, Hybrid GIPLs or LPG. Some pathways divert to free GPIs via side-chain modification rather than protein attachment.

Promastigotes express abundant GIPLs on their surface, with LPG and GPI-APs also present, while amastigotes rely more on GIPLs and lipids acquired from the host cell [56]. Genetic studies in *L. major* and *L. mexicana* show that deletion of LPG1 (galactofuranosyltransferase) prevents LPG synthesis but does not affect free GPI production, suggesting distinct biosynthetic branches [57]. Deletion of GPI biosynthetic enzymes like Dol-P-Man synthase is lethal unless rescued, demonstrating the essentiality of GPI products—especially free GPIs—for parasite viability [58]. In *Leishmania mexicana* mutants lacking GP18, GPI-anchored proteins (GPI-APs) were undetectable; however, the cells continued to synthesize free GPIs and LPG, maintained normal growth in culture, and remained infective to mice. This highlights the role of free GPIs in supporting parasite growth and infectivity [52]. Notably, the transition of *Leishmania major* from the promastigote stage in the sandfly vector to the intracellular amastigote stage within mammalian macrophages is accompanied by a marked downregulation of major surface macromolecules. In contrast, GIPLs, the predominant glycolipids of the parasite, are maintained at relatively constant levels throughout both developmental stages [59]. The diversity and stage-specific expression of GPI and its derivatives suggest that *Leishmania* has evolved a sophisticated system to adapt its surface architecture to different host environments. Free GPIs, especially GIPLs, may not only serve structural roles but also act as immune modulators and protective barriers in the hostile intracellular milieu.

## 8. Physiological and Pathological Roles of Free GPI in Protozoan Parasites

Free GPIs in protozoan parasites are not merely biosynthetic intermediates but active molecules with important physiological and pathogenic roles.

### 8.1. Immune Modulation

Free GPIs are potent modulators of the host immune response. In *Leishmania major*, GIPLs suppress nitric oxide (NO) production by macrophages, impairing leishmanicidal activity [60]. Mechanistically, GIPLs interfere with PKC signaling and iNOS induction, blocking transcriptional responses to IFN-γ and LPS. The lipid moiety—especially the alkyl-acyl glycerol structure—is essential for this immunosuppressive effect.

*T. gondii* GPIs stimulate TLR2/4 signaling, triggering inflammatory cytokines such as TNF-α and IL-12 [42,61]. This response is dependent on galectin-3 and varies by parasite strain and GPI side-chain composition [44]. These effects contribute to the systemic inflammation seen in acute toxoplasmosis.

*P. falciparum* free GPIs are central to malarial pathogenesis. They mimic host signaling molecules, activate PKC and NF-κB, and induce pro-inflammatory cytokines [49,50]. This leads to symptoms such as fever, cachexia, and cerebral complications. Notably, antibodies against *P. falciparum* GPI are protective, making them attractive targets for malaria vaccines.

In addition, highly purified GPIs from *T. cruzi* trypomastigotes potently activated murine macrophages, inducing strong production of TNF-α, IL-12, and nitric oxide, thereby eliciting a robust proinflammatory response [62]. Their stimulatory activity is attributed to specific structural features, including additional galactose residues and unsaturated fatty acids in the lipid moiety. Thus, *T. cruzi* GPIs serve as key modulators of innate immunity and help shape early host responses during Chagas disease.

### 8.2. Growth and Transmission

Free GPIs are also important for parasite growth and transmission. In *Leishmania mexicana*, deletion of Dol-P-Man synthase, which blocks both GPI-AP and free GPI biosynthesis, is lethal. On the other hand, mutants lacking GPI-APs by deletion of GPI-transamidase, but retaining free GPIs, remain viable and still infective [52], indicating a crucial role for free GPIs in parasite fitness. In *T. brucei*, mutants deficient in GPI-APs can grow in vitro but fail to grow in bloodstream form and to colonize the tsetse fly midgut [19], suggesting that free GPIs and/or GPI-APs are essential for in vivo survival.

## 9. Physiological and Pathological Roles of Free GPI in Mammals

Although free GPIs are less abundant in mammalian cells compared to protozoa, emerging evidence suggests they are biologically relevant and may contribute to both normal physiology and disease. Free GPIs have been detected in various mammalian cell lines and tissues using the T5-4E10 monoclonal antibody, which recognizes a GalNAc-modified GPI lacking protein attachment [4]. Tissue studies revealed that free GPIs are enriched in the brainstem, spinal cord, and epididymis, with lower levels in other tissues such as liver and kidney. These molecules may arise when GPI biosynthesis exceeds demand for protein anchoring or when GPI transamidase is defective.

### 9.1. Inflammation and Complement Dysregulation

In Paroxysmal Nocturnal Hemoglobinuria (PNH), a hematological disorder characterized by GPI-AP deficiency. Among the PNH due to PIGT mutations, free GPIs accumulate on the cell surface [63]. In PIGT-deficient PNH, GPI transamidation is blocked, resulting in the absence of CD55 and CD59 but continued surface expression of free GPIs. These unanchored GPIs activate the inflammasome, elevating IL-1β and IL-18 levels and contributing to systemic inflammation [64]. Moreover, excess free GPI has been proposed to enhance complement activation, although the precise mechanism remains unclear. One speculative possibility is that the EtNP groups on free GPIs, which terminate in nucleophilic amino groups (-NH_2_), could provide additional reactive sites on the membrane, facilitating complement deposition and activation. These findings highlight free GPI as a potential contributor to auto-inflammatory and complement-mediated disorders.

### 9.2. Blood Group Antigen

A novel physiological role for mammalian free GPI has emerged with the identification of the Emm blood group system. The Emm antigen corresponds to a structural epitope within the GPI anchor, and Emm-negative individuals lack this antigen because of biallelic loss-of-function mutations in the PIGG gene, which encodes an enzyme responsible for transferring an EtNP to Man2 during GPI biosynthesis [65,66]. As a result, anti-Emm antibodies are generated that recognizes GPI-bearing EtNP on Man2. The discovery of PIGG-related Emm negativity established that certain free GPI structures themselves can serve as blood group antigens, thereby expanding the biological and clinical relevance of GPI metabolism to transfusion medicine and immunohematology.

## 10. Conclusions and Perspectives

In this review, we describe the structure and biosynthesis of GPI in protozoan parasites and mammals. Although the sequence of enzymatic additions differs among organisms, the core pathway of GPI biosynthesis is conserved among organisms. Key steps such as GlcNAc-PI and GlcN-PI formation, mannosylation, and terminal EtNP addition are mediated by homologous enzyme families, and the fundamental GPI core structure (EtNP-Man_3_-GlcN-PI) is broadly preserved. However, there is substantial evolutionary divergence in the side-chain modification. Additionally, protozoan parasites such as *T. cruzi*, *T. gondii*, and *Leishmania* produce large amounts of free GPIs/GIPLs that exhibit extensive species-specific diversity in glycan side-chains, lipid chain species, and overall molecular complexity.

Free GPIs represent a biologically active class of glycolipids, whose significance extends far beyond their classical role in anchoring proteins to the membrane. In protozoan parasites such as *T. brucei*, *T. cruzi*, *Leishmania* spp., *T. gondii*, and *P. falciparum*, GPIs serve as immunomodulators, virulence factors, and essential structural components. Their abundance and molecular diversity reflect evolutionary adaptations that facilitate host invasion, immune evasion, and vector transmission. In mammals, free GPIs are less prominent but increasingly recognized as physiologically relevant. Pathological accumulation of free GPIs is observed in conditions such as PIGT-deficient PNH, where they contribute to inflammasome activation and systemic inflammation.

Despite growing interest, major gaps remain in our understanding. The full spectrum of mammalian free GPI structures and their biosynthetic regulation remain largely undefined. The physiological roles of free GPIs in healthy tissues and their possible involvement in metabolic or inflammatory diseases are also not well understood. To overcome such issues, developing better analytical tools, such as high-sensitivity antibodies and MS-based techniques, to identify and quantify free GPI species, is required in the future.

## Figures and Tables

**Figure 1 ijms-26-11592-f001:**
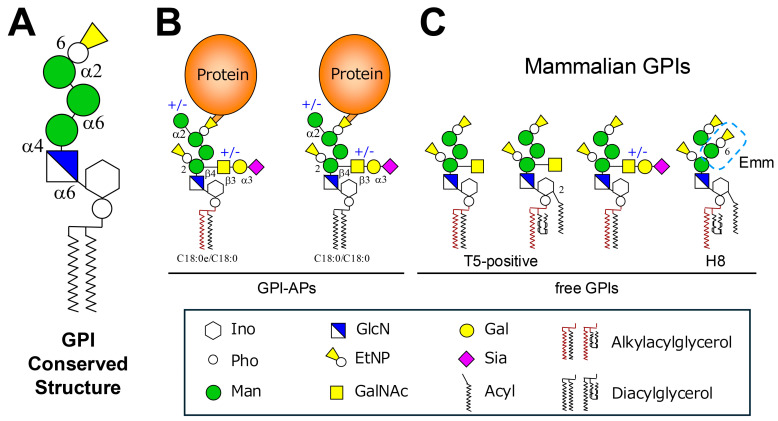
Structures of mammalian GPI-APs and free GPIs. (**A**) Conserved core glycan structure of GPI anchors among eukaryotes. The common GPI core consists of ethanolamine phosphate (EtNP) linked to the C-terminus of the protein, three mannose (Man) residues, and glucosamine (GlcN) attached to a inositol phospholipid moiety that varies among species. This conserved structure, represented as EtNP-6-Man-α1,2-Man-α1,6-Man-α1,4-GlcN-α1,6-myo-inositol-phospholipid, can be further modified by various side-chains. (**B**) Structure of mammalian GPI-APs. The GPI core glycan can be extended by an additional Man (Man4) and/or N-acetylgalactosamine (GalNAc) side-chains. The GalNAc branch can be elongated by galactose (Gal) and sialic acid (Sia) residues. The entire GPI-AP is attached to the outer leaflet of the plasma membrane via the hydrocarbon chains of phosphatidylinositol (PI), predominantly in the 1-alkyl-2-acylglycerol form, with diacylglycerol as a minor species. (**C**) Mammalian free GPIs. The left two structures correspond to T5-4E10 (T5)-reactive free GPIs, which contain a GalNAc side-chain and an EtNP group linked to the first Man. The inositol moiety may or may not be acylated. Addition of Gal residue to the GalNAc side-chain abolishes T5 reactivity (third structure from the left). The right structure represents a free GPI (H8), recognized as the Emm antigen, which carries EtNP on the second Man residues.

**Figure 2 ijms-26-11592-f002:**
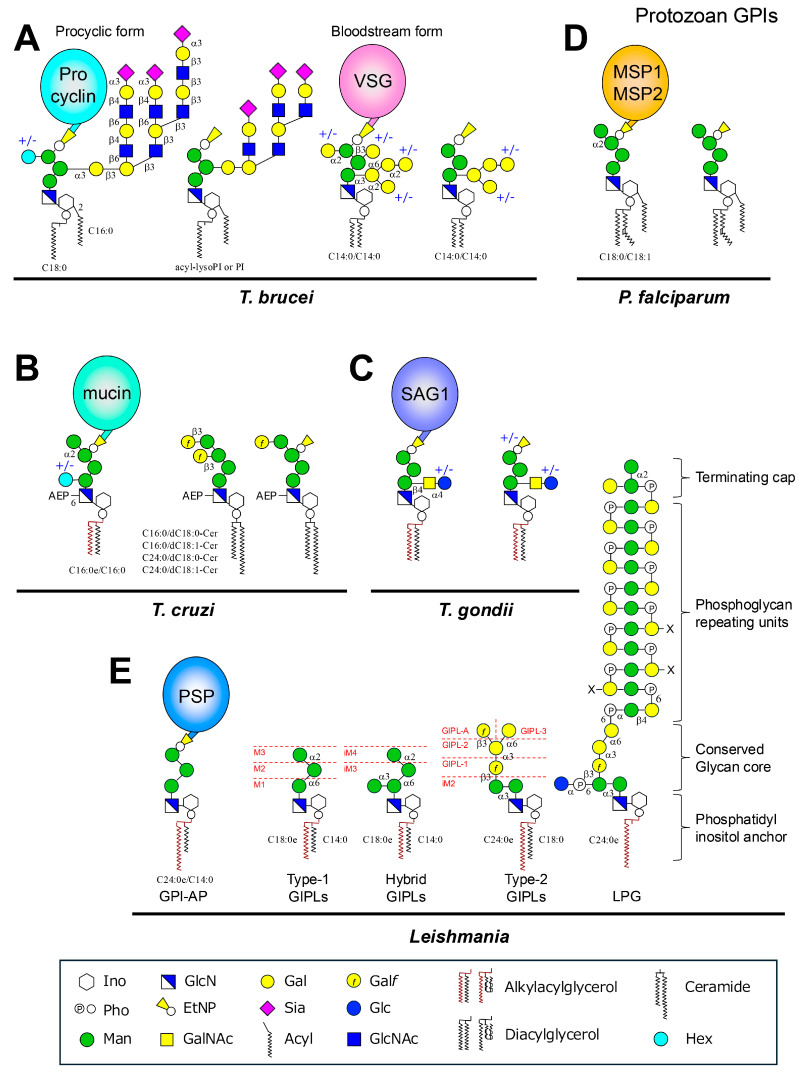
Schematic representations of GPI-APs and free GPIs in protozoan parasites. (**A**) *Trypanosoma brucei*: structures of the GPI-anchored surface glycoproteins procyclin (procyclic form) and VSG (bloodstream form), together with their corresponding free GPIs. (**B**) *Trypanosoma cruzi*: GPI-anchored mucin-like glycoprotein and its free GPI counterpart. (**C**) *Toxoplasma gondii*: GPI-anchored surface antigen SAG1 and the corresponding free GPI. (**D**) *Plasmodium falciparum*: GPI-anchored merozoite surface protein 1 (MSP1) and its free GPI. (**E**) *Leishmania* spp.: GPI-anchored protein PSP1, free GPIs (Type-1, Type-2, and Hybrid GIPLs), and lipophosphoglycan (LPG). Fatty acid composition in GIPLs, LPG, and GPI-APs varies among species. Dashed lines indicate truncated GIPL species. The number of phosphosaccharide repeats (*n*) in *Leishmania* LPGs is stage- and species-specific.

**Figure 3 ijms-26-11592-f003:**
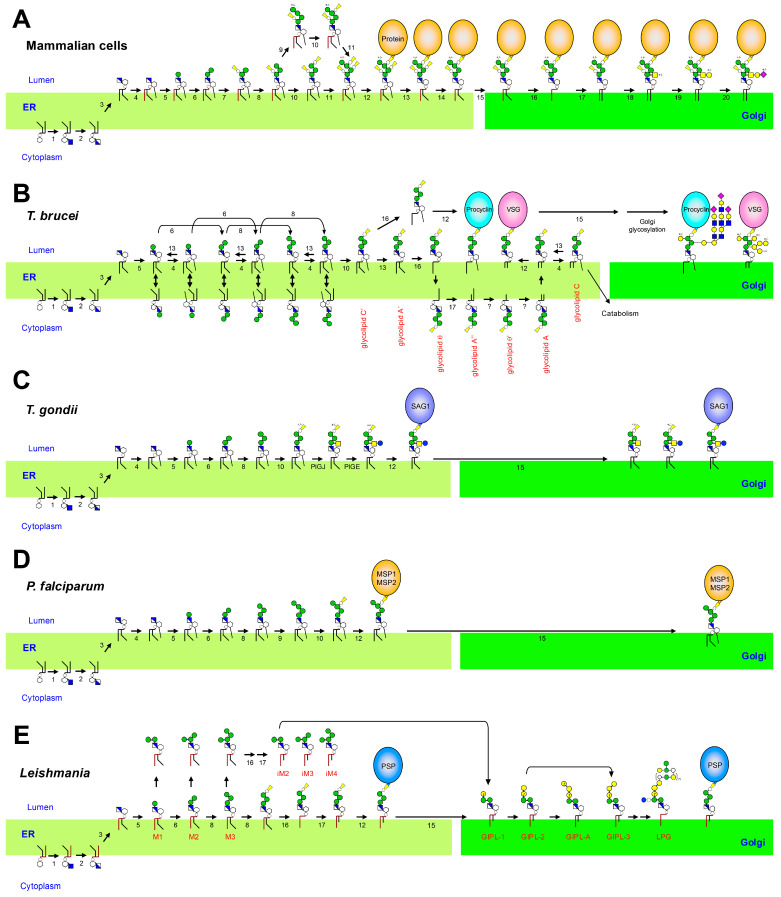
Biosynthetic pathways of GPIs in mammals and protozoa. (**A**) In mammalian cells, GPI biosynthesis occurs in the ER, followed by glycan and fatty acid remodeling in the Golgi apparatus before transfer to the cell surface. The first two steps take place on the cytoplasmic side of the ER, and subsequent sugar, lipid, and EtNP modifications occur on the luminal side. After attachment of the GPI anchor to proteins, further side-chain modifications may be introduced. The numbers indicated for each reaction in the figure correspond to those listed in Appendix A. (**B**) *Trypanosoma brucei*: GPI biosynthetic pathways in both bloodstream and procyclic forms. Fatty acid remodeling occurs prior to protein attachment in the ER. (**C**) *Toxoplasma gondii*: GPI biosynthesis involves species-specific side-chain additions catalyzed by PIGJ and PIGE glycosyltransferases. (**D**) *Plasmodium falciparum*: GPI biosynthesis retains the inositol-linked acyl chain without Golgi remodeling. (**E**) *Leishmania* spp.: GPI biosynthetic pathway producing both GPI-APs, GIPLs, and LPG.

## Data Availability

The original contributions presented in this study are included in the article/Appendix A. Further inquiries can be directed to the corresponding author.

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
