# Peer review of "Free GPIs and Comparison of GPI Structures Among Species"

_ijms, 2025, doi:10.3390/ijms262311592_

Round 1

Reviewer 1 Report

Comments and Suggestions for Authors

The authors have done an excellent job of covering a very complex field in an authoritative way.

General comments/suggestion:

  1. Section 3: Worth mentioning that the T. brucei transferrin receptor is GPI anchored as well as VSG. This may well be why GPI biosynthesis is essential in bloodstream form tryps in culture (no immune system pressure). See also section 3.3 statement about GPU essentiality.
  2. In section 3.3.1 might be worth mentioning that the inositol-deacylation step is essential for fatty acid remodelling to take place in bsf T. brucei
  3. In line 282 a copy number is provided of Toxo free GPIs – which is nice – but accurate copy numbers are available for most T. brucei, T. cruzi and leishmania GPI as well (but not quoted). Maybe where these numbers are available, and since average dimensions of the cells are known, the authors might think about expressing copy number per unit plasma membrane surface area to make comparisons?
  4. Although the immunomodulatory activities of Toxo and Plasmodium GPIs are described, the known proinflammatory activity of T. cruzi trypomastigote GPIs (with structure-activity relationships is not -see 2000 EMBO J paper by Almeida et al.
  5. Section 9.1 (and also relevant to discussions about Plasmodium GPI bioactivities. Does anyone really believe any of this insulin-mediator stuff anymore? The structure of IPGs was never tied down, a lot od reports used totally uncharacterised materials (where eg. mycoplasma contaminants may have been the active agent) - synthetic or highly purified and characterised GPIs do not have then purported IPG “insulin-mediator” activities – one does no favours to readers by referencing work that is likely flawed. The authors should consider this.
  6. Line 467-470: One idea/speculation, maybe the additional amino (nucleophile) groups on the outer plasma membrane thanks to the EtNP groups of the free GPIs is the reason for more complement activation?

Specific suggestions:

Line 83: ….is removed and replaced   would be better

Line 100: third Man should read first Man

Line 151 and elsewhere eg. line 155 and 233: procyclin GPI sidechains contain poly-N-acetyllactosamine (Galb1-4GlcNAc) extensions and poly lacto-N-biose (Galb1-3GlcNAc) extensions.

Line 156: and a proportion contained diacylglycerol – rather than phosphatidic acid

Line 216: Reacyltion with myristic acid

Line 234: exclusively Gal-based

Lines 319-320: I don’t think a reference is needed to say merozoites invade erythrocytes and if one did that would not be the key reference (ref 40)

Line 367-368: It consists of a lyso-alkyl-PI with a Gala1-6Gala1-3Galfb1-2(Glca1-P-6)Mana1-3Mana1-4GlcNa1-6 core glycan

Line 379-380: What is the evidence that “this heterogeneity impacts their immunogenicity and membrane behaviour~”? – maybe it does but I am not aware of any evidence for that.

Line 364: Ref 47 an odd choice to describe that leishmania family of GPIs (maybe use a comprehensive McConville review?)

Line 388: Promastigotes express abundant LPG and GPI-APs (true) but they express many more GIPLs than both of those.

Line 433: free GPIs and/or GPI-APs

Line 485; Free GPIs can serve as….. has a different meaning from GPIs serve as….. (not all of them do all those jobs)

Author Response

Comment 1: Section 3: Worth mentioning that the T. brucei transferrin receptor is GPI anchored as well as VSG. This may well be why GPI biosynthesis is essential in bloodstream form tryps in culture (no immune system pressure). See also section 3.3 statement about GPU essentiality.

Response 1: 

According to the reviewer’s suggestion, we added the sentences in Section 3.1:

“The bloodstream forms of T. brucei require iron for growth, which is delivered by the transferrin of the host. The transferrin receptor of T. brucei is known to be different in structure, subunit organization, and mode of membrane anchorage from that of the mammalian host (Trevor et al., 2019. Nat Microbiol., 4:2074-2081). The trypanosome transferrin receptor, which is a heterodimer of related proteins ESAG6 and ESAG7, is attached to the cell membrane by a single GPI anchor on ESAG6 (Urbaniak and Gadelha, 2023. Cell Surf., 9:100100).” (Page 4, Line 144-150)

In addition, we added the sentences in Section 3.3:

“In bloodstream form T. brucei, TbGPI10 is essential, and its loss is lethal because GPI-anchored VSGs are required for survival, and probably in addition to the GPI-anchored transferrin receptor required for acquiring iron and viability.” (Page 7, Line 190-191)

Comment 2: In section 3.3.1 might be worth mentioning that the inositol-deacylation step is essential for fatty acid remodelling to take place in bsf T. brucei.

Response 2: According to the reviewer’s suggestion, we added the information in Section 3.3.1:

“In the bloodstream form, unlike the procyclic forms, GPI inositol-deacylation is essential to complete fatty acid remodeling.” (Page 7, Line 203-204)

Comment 3: In line 282 a copy number is provided of Toxo free GPIs - which is nice - but accurate copy numbers are available for most T. brucei, T. cruzi and leishmania GPI as well (but not quoted). Maybe where these numbers are available, and since average dimensions of the cells are known, the authors might think about expressing copy number per unit plasma membrane surface area to make comparisons?   

Response 3: 

We thank the reviewer for this insightful suggestion. We agree that expressing GPI abundance relative to plasma membrane surface area would, in principle, provide a more meaningful basis for comparing GPI density across different parasites. Accordingly, we examined the available quantitative data on GPI copy numbers in T. brucei, T. cruzi, Leishmania, and T. gondii.

T. brucei: Approximately 1 × 107 copies of the VSG are expressed on the surface of bloodstream-form parasites, accounting for ~10% of total protein (Allison et al., 2014. Microb Cell 1:325-345) (Guha-Niyogi et al., 2001. Glycobio 11:45R-59R) (Page 4, Line 143-144)

T. cruzi: The estimated free GPI copy number is approximately 1.5 × 107 molecules per cell (Lederkremer et al., 1991. J Biol Chem 266:23670-23675) and ~2–4 × 106 copies of GPI-anchored mucins are expressed per parasite (Canepa et al., 2012. J Biol Chem 287:26365-26376). (Page 8, Line 269-271)

Leishmania: GIPLs and LPG are present at approximately 1 × 107 and ~5 × 106 molecules per cell, respectively (Forestier et al., 2014. Front Cell Infect Microbiol 4:193). (Page 11, Line 372-373)

While these values are useful for comparison, reported GPI amounts are generally based on indirect estimates rather than precise molecular counts. In addition, parasite surface area cannot be determined accurately due to variability in cell morphology. Thus, calculating GPI abundance per surface area would require multiple assumptions and could give a false sense of precision. For these reasons, we chose not to include normalized values by surface-area and instead provide a brief comparative summary of reported GPI copy numbers.

Comment 4: Although the immunomodulatory activities of Toxo and Plasmodium GPIs are described, the known proinflammatory activity of T. cruzi trypomastigote GPIs (with structure-activity relationships is not -see 2000 EMBO J paper by Almeida et al.

Response 4: 

Thank you for the helpful suggestion. We updated Section 8.1 in the revised manuscript:

“In addition, highly purified GPIs from T. cruzi trypomastigotes potently activated murine macrophages, inducing strong production of TNF‑α, IL‑12, nitric oxide, thereby eliciting a robust proinflammatory response (Almeida et al., 2000. Embo J., 19:1476-1485). Their stimulatory activity is attributed to specific structural features, including additional galactose residues and unsaturated fatty acids in the lipid moiety. Thus, T. cruzi GPIs serve as key modulators of innate immunity and help shape early host responses during Chagas disease.” (Page 12, Line 439-444)

Comment 5: Section 9.1 (and also relevant to discussions about Plasmodium GPI bioactivities. Does anyone really believe any of this insulin-mediator stuff anymore? The structure of IPGs was never tied down, a lot of reports used totally uncharacterised materials (where eg. mycoplasma contaminants may have been the active agent) - synthetic or highly purified and characterised GPIs do not have then purported IPG "insulin-mediator" activities - one does no favours to readers by referencing work that is likely flawed. The authors should consider this.

Response 5: 

According to the reviewer’s suggestion, we removed the IPGs part in Section 6.3 and 9.1.

Comment 6: Line 467-470: One idea/speculation, maybe the additional amino (nucleophile) groups on the outer plasma membrane, thanks to the EtNP groups of the free GPIs is the reason for more complement activation? 

Response 6: 

Thank you for the idea. We added the following sentences in Section 9.1 of the revised manuscript.

“One speculative possibility is that the ethanolamine phosphate (EtNP) groups on free GPIs, which terminate in nucleophilic amino groups (-NHâ‚‚), could provide additional reactive sites on the membrane, facilitating complement deposition and activation.” (Page 13, Line 469-472)

Other specific suggestions

a. Line 83: ….is removed and replaced would be better.

We revised the manuscript. (Page 2, Line 82)

b. Line 100: third Man should read first Man.

This has been corrected to the first Man. Thank you. (Page 3, Line 99)

c. Line 151 and elsewhere eg. line 155 and 233: procyclin GPI sidechains contain poly-N-acetyllactosamine (Galb1-4GlcNAc) extensions and poly lacto-N-biose (Galb1-3GlcNAc) extensions.

This has been modified and corrected accordingly. (Page 4, Line 156-157 and Page 8, Line 243)

d. Line 156: and a proportion contained diacylglycerol – rather than phosphatidic acid.

This has been corrected accordingly. (Page 4, Line 162)

e. Line 216: Reacyltion with myristic acid.

This has been corrected accordingly. (Page 7, Line 223)

f. Line 234: exclusively Gal-based.

This has been modified and corrected accordingly. (Page 8, Line 241)

g. Lines 319-320: I don’t think a reference is needed to say merozoites invade erythrocytes and if one did that would not be the key reference (ref 40)

According to the suggestion, the reference was deleted.

h. Lines 367-368: It consists of a lyso-alkyl-PI with a Gala1-6Gala1-3Galfb1-2(Glca1-P-6)Mana1-3Mana1-4GlcNa1-6 core glycan.

According to the suggestion, we corrected the manuscript. (Page 11, Line 376-378)

i. Line 379-380: What is the evidence that “this heterogeneity impacts their immunogenicity and membrane behaviour~”? – maybe it does but I am not aware of any evidence for that.

The structural heterogeneity of GIPLs, in both glycan and lipid moieties, may contribute to differences in immune recognition. We agree that direct evidence demonstrating a causal link between specific GIPL types and defined “membrane behaviors” is limited. We revised the manuscript as follow:

“This heterogeneity might impact their immunogenicity. For example, GIPLs from L. braziliensis (galactose-rich, Type II) were shown to more strongly inhibit nitric oxide and IL-12 production in primed macrophages compared with GIPLs from L. infantum (Type I/hybrid) (Assis et al., 2012. PLoS Negl Trop Dis., e1543), indicating that structural differences can modulate macrophage responses.” (Page 11, Line 390-393)

j. Line 364: Ref 47 an odd choice to describe that leishmania family of GPIs (maybe use a comprehensive McConville review?)

A new reference by McConville was added (McConville et al., 1990. J Biol Chem 265:7385-7394). (Page 11, Line 372)

k. Line 388: Promastigotes express abundant LPG and GPI-APs (true) but they express many more GIPLs than both of those.

This has been modified in section 7.3 as follows:

“Promastigotes express abundant GIPLs on their surface, with LPG and GPI-APs also present.” (Page 11, Line 401-402)

l. Line 433: free GPIs and/or GPI-APs.

This has been corrected accordingly. (Page 12, Line 452)

m. Line 485; Free GPIs can serve as….. has a different meaning from GPIs serve as….. (not all of them do all those jobs).

This has been corrected accordingly. (Page 13, Line 497)

Reviewer 2 Report

Comments and Suggestions for Authors

The review compares the biosynthesis and functional roles of GPI-AP and free GPI in protozoa and mammals. It draws attention to differences in Trypanosoma brucei, T. cruzi, Toxoplasma gondii, Plasmodium falciparum, Leishmania spp., and mammals in their GPI anchor biosynthesis pathway. Additionally, highlighting how free GPI are distinct from GPI-AP in terms of their structure. With emphasis on their role in host-pathogen interactions, protozoans’ viability, and immune response, the review focuses on their physiological and pathological functions, their abundance in cell, and stage-specific functions of free GPI in protozoans. Also, the free GPI in P. falciparum are mentioned as potential targets for vaccination development. In mammals, the distribution of free GPI in mammalian tissues and their function in immune response are described. It also describes the physiological significance of free GPI, such as insulin signaling and how immune mediated response is triggered on absence Emm antigen on red blood cells. Overall, the review summarizes the importance of free GPI in the physiological and pathological processes, host-pathogen interaction, and also potential therapeutic uses.

Minor comments:

  1. Is there any literature currently available on the molecular mechanisms by which free GPIs regulates immune response and host-parasite interactions?
  2. Have the mechanisms regulating the free GPI biosynthesis its modifications, been characterized across protozoan and mammals, especially in Leishmania and mammals as free-GPI in these two are shown to have more modifications?
  3. What is known regarding the evolutionary conservation or divergence of free GPI biosynthesis among protozoans and mammals?
  4. Additionally, there are minor corrections in the manuscript like falciparum should have been italicized in line 424, and “Free” GPI should have been corrected to “free” GPI with “f” in lowercase in line 444.
  5. In the table, the cerevisiae homolog of mammalian DPM2 is not mentioned. Yil102c-A has been identified as DPM2 in S. cerevisiae (Pilsyk S, et al. 2020).

Author Response

Comment 1: Is there any literature currently available on the molecular mechanisms by which free GPIs regulates immune response and host-parasite interactions?

Response 1: 

Yes, there is some literature describing molecular mechanisms through which free GPIs regulate immune responses. For example, T. cruzi–derived GPI anchors and GIPLs (Campos et al. 2001.J Immunol 167:416-423) have been shown to induce IL-12, TNF-α, and nitric oxide production in macrophages in a TLR2-dependent manner, indicating that T. cruzi GPIs are potent activators of TLR2. Such TLR2 activation is thought to initiate innate immune defenses and inflammatory responses during protozoan infection.

In P. falciparum, GPIs act as potent pathogen-associated molecular patterns (PAMPs), acti-vating innate immune receptors (e.g., TLR2 and TLR4) and stimulating production of pro-inflammatory cytokines such as TNF-α and IL-1β (Mbengue et al., 2016. Immun Inflamm Dis 4:24-34) (Mockenhaupt et al., 2006. Proc Natl Acad Sci USA 103:177-182).

However, despite these findings, the detailed molecular mechanisms by which free GPIs modulate immune responses and host–parasite interactions remain incompletely understood.

We have incorporated these points into the revised manuscript. (Page 12, Line 439-444; Page 10, Line 351-355)

Comment2: Have the mechanisms regulating the free GPI biosynthesis its modifications, been characterized across protozoan and mammals, especially in Leishmania and mammals as free-GPI in these two are shown to have more modifications?

Response 2:

Yes, there is some literature showing regulators of free GPI biosynthesis. In mammalian cells, accumulated precursors of specific GPI-anchored proteins can upregulate GPI biosynthesis in an ARV1-dependent manner (Liu et al., 2023. J Cell Biol 222). We added these points to the revised manuscript. (Page 4, Line 122-124)

Comment 3: What is known regarding the evolutionary conservation or divergence of free GPI biosynthesis among protozoans and mammals?

Response 3:

Thank you for this important question. Although the sequence of enzymatic additions differs among organisms, the core pathway of GPI biosynthesis is conserved among organisms. Key steps such as GlcNAc-PI and GlcN-PI formation, mannosylation, and terminal EtNP addition are mediated by homologous enzyme families, and the fundamental GPI core structure (EtNP-Man₃-GlcN-PI) is broadly preserved. However, there is substantial evolutionary divergence in the side-chain modification. Additionally, protozoan parasites such as T. cruzi, T. gondii, and Leishmania produce large amounts of free GPIs/GIPLs that exhibit extensive species-specific diversity in glycan side-chains, lipid chain species, and overall molecular complexity.

We have included these points in the revised manuscript. (Page 13, Line 484-494)

Comment 4: Additionally, there are minor corrections in the manuscript like falciparum should have been italicized in line 424, and “Free” GPI should have been corrected to “free” GPI with “f” in lowercase in line 444.

Response 4:

This has been corrected accordingly in the manuscript. (Page 12 , Line 443)

Comment 5: In the table, the cerevisiae homolog of mammalian DPM2 is not mentioned. Yil102c-A has been identified as DPM2 in S. cerevisiae (Pilsyk S, et al. 2020).

Response 5:

This has been corrected accordingly in the supplementary table for the manuscript.
